# Genome-Wide Identification and Characterization of *HSP70* Gene Family in Tausch’s Goatgrass (*Aegilops tauschii*)

**DOI:** 10.3390/genes16010019

**Published:** 2024-12-26

**Authors:** Yongmei Xu, Yue Liu, Yanjun Yi, Jiajia Liu

**Affiliations:** 1College of Life Science, Qingdao Agricultural University, Qingdao 266109, China; xym@stu.qau.edu.cn; 2College of Horticulture, Qingdao Agricultural University, Qingdao 266109, China; 3Haidu College Qingdao Agricultural University, Qingdao 266603, China

**Keywords:** *HSP70* gene family, molecular characteristics, genome-wide, weed, *Aegilops tauschii*

## Abstract

Background: *Aegilops tauschii*, a winter annual grass weed native to Eastern Europe and Western Asia, has become a widespread invasive species in the wheat-growing regions of China due to its high environmental adaptability. This study aims to explore the molecular mechanisms underlying the stress resistance of Tausch’s goatgrass, focusing on the *HSP70* gene family. Methods: A genome-wide analysis was conducted to identify and characterize the *HSP70* gene family in *A. tauschii*. Afterward, their physicochemical properties, phylogenetic relationships, gene structures, and chromosomal distributions were analyzed. Additionally, cis-acting regulatory elements were predicted to understand their potential role in stress resistance. Results: A total of 19 identified *HSP70* family genes were classified into four subfamilies and distributed across all chromosomes. The syntenic analysis revealed extensive homology between Tausch’s goatgrass and wheat *HSP70* genes. Segmental duplication was found to play a crucial role in the expansion of the *HSP70* gene family. The prediction of cis-acting elements suggested that these genes are involved in stress resistance to various environmental conditions. Conclusions: This study provides a comprehensive overview of the *HSP70* gene family in *A. tauschii*, offering insights into their role in stress resistance and their potential application in understanding invasive species behavior and improving wheat resilience. Further research is needed to validate their functional roles in stress adaptation.

## 1. Introduction

Heat shock proteins (Hsps) are a conserved superfamily of molecular chaperones that play crucial roles in maintaining cellular protein homeostasis, particularly under stress conditions such as heat [1] and drought [2], cold [3]), salinity [4], and heavy metals [5]. Based on their molecular weight, Hsps are classified into several major families, including Hsp100, Hsp90, Hsp70, Hsp60, and small Hsp (<30 kDa) [6,7]. Among these, the 70-kDa heat shock proteins (Hsp70s) are one of the most conserved and extensively studied chaperone families across eukaryotes and prokaryotes organisms [8]. The Hsp70 proteins typically consist of two key functional domains: an amino (N)-terminal ATPase domain and a carboxyl (C)-terminal peptide-binding domain [9]. These domains allow Hsp70 to facilitate the folding, refolding, and transport of nascent and denatured proteins, ensuring proper protein conformation and preventing irreversible aggregation under stress [8,10,11]. By participating in protein quality control and cellular stress responses, Hsp70s act as essential regulators of protein homeostasis, playing critical roles in plant adaptation to adverse environmental conditions [8,11,12].

*A. tauschii* (Tausch’s goatgrass), the donor of the D-genome in hexaploidy bread wheat (*Triticum aestivum* L.), played a pivotal role in the evolution of modern bread wheat through hybridization with tetraploid emmer wheat approximately 8000 years ago [13,14,15]. Geographically, *A. tauschii* is predominantly distributed across the Mediterranean shoreline, Southern Europe, Northern Africa, the Middle East, and Southwest Asia [16]. Over time, its range has expanded to other regions, including China, where it has become a problematic invasive species in wheat fields due to its exceptional environmental adaptability [17]. *A. tauschii* demonstrates notable resilience to abiotic stresses, including reduced osmotic potential, elevated salinity, and extreme temperature fluctuations, which enables it to thrive in diverse environments [18]. This high adaptability makes *A. tauschii* not only a valuable resource for studying stress tolerance mechanisms but also a competitive weed that impacts wheat productivity by competing for essential resources such as nutrients, water, and light [19,20,21]. Given its ecological significance, understanding the molecular mechanisms underlying *A. tauschii*’s stress resilience is essential for both agricultural weed management and the improvement of wheat stress tolerance.

Although the *HSP70* gene family has been widely characterized in many plants, including crops, little is known about its roles in stress tolerance and environmental adaptability in *A. tauschii*. Given the ecological significance of *A. tauschii* as a highly adaptable wild wheat relative, understanding the molecular mechanisms underlying its stress resilience is essential. The identification and characterization of the *HSP70* gene family in *A. tauschii* may provide new insights into its ability to withstand harsh environmental conditions, which can further inform strategies to mitigate its spread and improve wheat stress resistance.

While the *HSP70* gene family has been extensively characterized in several plant species [22,23,24], limited information is available regarding its diversity, structure, and function in *A. tauschii*. In this study, we performed a comprehensive genome-wide identification and characterization of the *HSP70* gene family in *A. tauschii*. Specifically, we analyzed their chromosomal locations, gene structures, phylogenetic relationships, conserved domains, synteny, and cis-acting regulatory elements. This study aims to provide a foundation for further functional investigations into the roles of Hsp70s in stress tolerance and their potential applications in agriculture. Understanding the Hsp70 family in *A. tauschii* may not only reveal key mechanisms of environmental adaptability but also contribute to developing strategies for enhancing stress resilience in wheat and other crops.

## 2. Materials and Methods

### 2.1. A. tauschii Characteristics

*A. tauschii*, an annual grass species belonging to the Poaceae family, is considered a weed. Propagated through seeds, *A. tauschii* exhibits robust tillering, averaging 10–20 tillers per plant, with the maximum exceeding 32. The growth cycle of *A. tauschii* features two germination peaks: one in the autumn, 15–20 days after wheat sowing, and another in the spring, from late February to March of the following year. *A. tauschii* matures 5–7 days earlier than wheat, shedding its rachis as it matures, leaving only the base’s 1–2 nodes [19]. Originating in Western Asia, *A. tauschii* is distributed across various provinces in China, demonstrating strong adaptability and the ability to grow in arid and saline-alkali soils. *A. tauschii* has a broad distribution, spanning the Eurasian continent, with Iran recognized as its center of genetic diversity and origin. In China, *A. tauschii* is primarily concentrated in the Xinjiang Ili region and the Yellow River basin [25].

### 2.2. Genome-Wide Identification of Hsp70 Family Members

Data resources of genome, proteome, and annotation of *A. tauschii* (Aet_v4.0 assembly) and *Triticum aestivum* (IWGSC assembly) were obtained from the EnsemblPlants (https://plants.ensembl.org, accessed on 3 April 2024). To identify the possible Hsp70 homologies in the whole genome, both the local application of BLAST for local sequence alignment and HMM for probabilistic modeling were employed [26]. The reviewed Hsp70 amino acid sequences of Arabidopsis (*Arabidopsis thaliana*) and rice (*Oryza sativa*) (Appendix A) were obtained from the TAIR (http://www.arabidopsis.org/, accessed on 5 May 2024) and UniProt (https://www.uniprot.org/, accessed on 5 May 2024) database, which were taken as a query to search for the potential Hsp70 sequences in *A. tauschii* via BLAST with a threshold of e-value = 1 × 10^−10^. In addition, the potential Hsp70 protein sequences were searched against the HMM profile of the Hsp70 domain (PF00012), downloaded from the Pfam (http://pfam-legacy.xfam.org/, accessed on 9 May 2024), via HMMER v3.2 (http://hmmer.org/, accessed on 6 June 2024) with the default setting. The protein sequences were identified as Hsp70 candidates with both BLAST and HMM approaches. After the removal of redundant sequences, the Hsp70 conserved domain was confirmed in the candidate protein sequences through SMART (http://smart.embl-heidelberg.de/, accessed on 10 June 2024).

### 2.3. Physicochemical Properties Analysis of Hsp70 Family Members

The identified non-redundant Hsp70 family protein information, including the length of the amino acid sequence, genomic sequences, and full CDS sequences, were extracted from the genome annotation file using TBtools v1.0 [27]. The molecular mass, isoelectric pH, and grand average hydropathy index, which are key physicochemical parameters for each Hsp70 protein sequence, were calculated via ExPASy (https://web.expasy.org/protparam/, accessed on 6 July 2024). The intracellular distribution of all identified Hsp70 proteins was predicted using the deep learning method in the DeepLoc v2.0 program [28].

### 2.4. Multiple Alignments and Phylogenetic Analysis of Hsp70 Proteins

The identified Hsp70 protein sequences of *A. tauschii* (AetHsp70) and wheat (TaHsp70), together with known Hsp70 in Arabidopsis (AtHsp70) and rice (OsHsp70), were aligned using ClusalW with default parameters. Phylogenetic trees were reconstructed using the neighbor-joining (NJ) method in the MEGAX program [29,30], and bootstrap tests were implemented using 1000 replications.

### 2.5. Gene Structure Analysis and Identification of Conserved Motifs

In order to delve into the variegated composition and structural intricacies of the group’s constituents, a thorough examination of Hsp70 was conducted to elucidate the diversity and structural intricacies within the gene family. We conducted a comparative analysis of exon-intron configurations to analyze the exon-intron structure of genes. We utilized the Gene Structure Display Server (GSDS) 2.0 (https://gsds.gao-lab.org/index.php, accessed on 25 July 2024), which facilitates the visualization of gene features such as exons, introns, and conserved elements [31], and visualized in TBtools v1.0 [27]. The conserved motifs were detected in MEME 5.0 [32,33]. The limits on minimum width, maximum width, and maximum counts for the identified motifs were designated as 6, 50, and 10, respectively.

### 2.6. Chromosomal Location and Syntenic Analysis

Positional information of predicted *HSP70* genes was extracted from each genomic sequence and annotation file and then was visualized in TBtools v1.0 [27]. The identified Hsp70s were mapped to the chromosomes of *A. tauschii* and *T. aestivum*. Genomic comparisons were determined by all-against-all BLASTP searches (e-value = 1 × 10^−10^) using the proteome sequences of *A. tauschii* and *T. aestivum* as queries against these two species proteomes. Syntenic analysis between *A. tauschii* and *T. aestivum* for homologous *HSP70* family genes was conducted using the MCScanX toolkit [34]. Meanwhile, the collinear of *HSP70* family genes in *A. tauschii* and *T. aestivum* were tested. The syntenic analysis results were visualized using TBtools v1.0 [27].

### 2.7. Cis-Acting Element Prediction in Promoters

To assess the potential regulatory impact of various cis-acting elements within the promoter regions, we conducted a detailed analysis. *AetHSP70* genes and promoter sequences within 2000 bp upstream were investigated from the genomic sequences, and the *cis*-acting regulatory elements in the regions were explored using the PlantCARE (http://nfix2008.psb.ugent.be/webtools/plantcare/html/, accessed on 20 August 2024) database [35]. Hence, the number of *AetHsp70s* contained in each of the predicted *cis*-acting elements was calculated.

## 3. Results

### 3.1. Identification of Hsp70 Family Members in A. tauschii

To extensively identify Hsp70 family members in *A. tauschii*, we performed a genome-wide scan using both BLAST and HMM profile searches. After checking the conserved domain in Hsp70 proteins using the SMART program, a total of 19 *HSP70* family genes named *AetHSP70-1* to *AetHSP70-19* were identified in *A. tauschii* genome (Table 1). In parallel, 58 *TaHSP70s* (*TaHSP70-1* to *TaHSP70-58*) were recognized in *T. aestivum* (Appendix A), which was approximately three times of *AetHSP70s*. Analyses of the physiological properties showed that the *AetH70s* encoding proteins consist of 575 (*AetHSP70-14*) and 737 (*AetHSP70-7*) amino acids. The molecular weights of AetHsp70s were between 61.89 kDa (*AetHSP70-14*) and 81.13 kDa (*AetHSP70-7*). Most of the AetHsp70 proteins, with the exception of one, demonstrated stability in vitro, and their isoelectric points (pI) were consistently determined. AetHsp70-7 (pI = 7.05) had low isoelectric points (pI < 7). The molecular weights of AetHsp70s were between 61.89 kDa (*AetHSP70-14*) and 81.13 kDa (*AetHSP70-7*). Most of the AetHsp70 proteins, with the exception of one, demonstrated stability in vitro, and their isoelectric points (pI) were consistently determined. AetHsp70-7 (pI = 7.05) had low isoelectric points (pI < 7). The molecular weights of AetHsp70s were between 61.89 kDa (*AetHSP70-14*) and 81.13 kDa (*AetHSP70-7*). Most of the AetHsp70 proteins, with the exception of one, demonstrated stability in vitro, and their isoelectric points (pI) were consistently determined. AetHsp70-7 (pI = 7.05) had low isoelectric points (pI < 7). The GRAVY value of all AetHsp70s was negative (−0.54–2.1), indicating that the AetHsp70 proteins were hydrophilic and suggesting that AetHsp70s might be possibly involved in tolerance to drought stress [36]. Subcellular localization prediction showed that the AetHsp70 proteins were differently located on the endoplasmic reticulum (6), followed by plastid (5), cytoplasm (4), and mitochondrion (4). 

### 3.2. Phylogenetic Analysis of the Hsp70 Family Proteins

To assess the phylogenetic relationships and evolutionary pattern of Hsp70 family proteins, we conducted a phylogenetic analysis of 19 AetHsp70s in *A. tauschii*, 58 TaHsp70s in *T. aestivum*, 17 AtHsp70s in *A. thaliana*, and 8 OsHsp70s in *O. sativa*. A total of 102 Hsp70 amino acid sequences were aligned to generate an NJ tree (Figure 1). As a result, both phylogenetic analyses revealed that similar topologies and evolutionary structures partitioned the Hsp70 proteins into four major clades: Hsp70-I, Hsp70-II, Hsp70-III, and Hsp70-IV subfamilies. The subfamily Hsp70-III, the largest subfamily, encompassed 32 members, whereas the subfamily Hsp70-IV had 18. The members likely to be truncated were identified by aligning with the genomic DNA sequences of A. thaliana and *O. sativa* orthologs [37]. Meanwhile, the subfamily Hsp70-I comprised 28 members, and the subfamily Hsp70-II contained 24. The analysis of subcellular localization predictions indicated that the Hsp70 proteins encoded by the genes within the subfamily are likely to be localized in various cellular compartments. Hsp70-II was located in the endoplasmic reticulum and plastid. Cytoplasmic and mitochondrial *HSP70* genes mainly clustered on the subfamilies Hsp70-I and Hsp70-III. In addition, the Hsp70 proteins of *A. thaliana* and *O. sativa* were found across all subfamilies. This distribution also suggested that each subfamily contained proteins with diverse cellular roles. *AetHSP70* genes had orthologs in the genome of *A. thaliana* and *O. sativa*.

### 3.3. Structure of AetHSP70 Genes and Conserved Motifs of AetHsp70 Proteins

To explore the structural characteristics of the *AetHSP70* gene family during its evolution, the conserved motifs on each AetHsp70 protein and the exon–intron organization of individual *AetHSP70* genes were projected (Figure 2A). As the results have shown, a total of 10 conserved motifs were recognized, with the length of amino acids ranging from 29 to 50 amino acids (Figure 2B). The ten motifs were conserved and found in all AetHsp70 protein sequences except AetHsp70-14 and AetH70-15. Both the AetHsp70-14 and AetHsp70-15 belong to the Hsp70-IV subfamily. According to the amino acid sequence composition, motif 9 was absent in AetHsp70-14, and motif 8 was lost in AetHsp70-15. Consequently, the highly conserved protein structure of AetHspsp70s indicates their similar functions in *A. tauschii*. While some motif sequences changed slightly or the minority was lost, these features potentially enhanced their specialized biological roles.

Prediction of gene structure in *AetHSP70s* presented that the arrangement of exons and introns in the whole *HSP70* gene family was complex. There were eight exons in *AetHSP70-9*, *AetHSP70-14*, and *AetHSP70-15*, but only one exon was found in *AetHSP70-13*. In addition, the number of introns in total genes ranged from 0 (*AetHSP70-10, AetHSP70-11*, and *AetHSP70-13*) to 7 (*AetHSP70-15* and *AetHSP70-15*). Some of the *AetHSP70s* had fewer intron numbers displaying a longer exon phase. Based on the nucleotide sequence composition, we could find that some intron loss and gain events may have occurred during the structural evolution among the *AetHSP70* genes in *A. tauschii*. Moreover, the dimensions and arrangement of the 3′ and 5′ untranslated regions (UTRs) exhibited diversity within the non-coding regions. Gene structural analysis revealed that while the structure of introns and UTRs varied significantly, the essential coding sequences remained consistent across all nucleotide sequences of *HSP7* family genes in *A. tauschii*.

### 3.4. Chromosomal Distribution and Syntenic Analysis of HSP70 Genes

The chromosomal location of all identified *HSP70* genes in *A. tauschii* and *T. aestivum* was considered based on the physical position of whole genes (Figure 3). Nineteen *AetHSP70s* were distributed across all chromosomes of *A. tauschii,* but the number of genes on each chromosome varied considerably. For the *A. tauschii*, chromosome 5D carried five *AetHSP70s*, including *AetHSP70-2*, *AetHSP70-15*, *AetHSP70-16*, *AetHSP70-11*, and *AetHSP70-4*. Then, chromosome 4D carried four *AetHSP70s*, and five other chromosomes had two *AetHSP70s*. For the *T. aestivum*, 58 *TaHSP70s* had non-random distribution across all chromosomes, with 19 in the D-genome, 17 in the A-genome, 21 in the B-genome, and 1 in the unmapped chromosome. The distribution of 19 *TaHSP70* genes in the D-genome was identical to the orthologous *AetHSP70* genes of *A. tauschii*. However, the *AetHSP70-18* at 524.59 Mb on chromosome 4D has not been retained in *T. aestivum*; it may suggest that this gene was lost during allopolyploidization. The chromosomal positioning of *HSP70s* appears to have originated from extensive gene duplication events throughout evolutionary history.

To better understand the *HSP70* gene family expansion and clustering, a comparative analysis of syntenic gene maps was performed in *A. tauschii* and *T. aestivum*. Between these two species, a total of 112,475 collinear genes were detected, which occupied 76.87% of whole genes and covered almost all chromosomes (Figure 4A). Among the *HSP70* family genes, 17 *AetHSP70s* were shown to be syntenic with 43 *TaHSP70s*. Out of them, *AetHSP70-19* (2D), *AetHSP70-15* (5D), and *AetHSP70-13* (7D) were collinear with *TaHSP70-58* (2D), *TaHSP70-46* (5D), and *TaHSP70-39* (7D), respectively. Likewise, *AetHSP70-6* of *A. tauschii* was discovered to be collinear with only *TaHSP70-21* on chromosome 3B in wheat. With the exception of these four *AetHSP70s*, the other *AetHSP70s* were not only collinear with *TaHSP70s* on the D-genome but also showed collinearity with *TaHSP70s* on the A-genome and/or B-genome. For example, *AetHSP70-14* on chromosome 1D was separately syntenic to *TaHSP70-43* on chromosome 1A, *TaHSP70-44* on chromosome 1B, and *TaHSP70-45* on chromosome 1D.

We further identified three pairs of syntenic *HSP70* genes in *A. tauschii* (Figure 4B) and 49 pairs of ones in *T. aestivum* (Figure 4C). In order to understand the expansion mechanism of the *HSP70s*, the gene duplication events (singleton, tandem, and segmental duplications) were investigated. Among all the *AetHSP70s*, *AetHSP70-2*, and *AetHSP70-3*, *AetHSP70-6*, *AetHSP70-7*, *AetHSP70-6*, and *AetHSP70-8* were separately located in different collinear gene blocks as segmental duplications. Except for these five *AetHSP70s*, there were 11 *HSP70* family members who were singleton without duplication in the *A. tauschii* genome. Compared to *AetHSP70s*, the syntenic pattern of *TaHSP70s* was more complex. One tandem *TaHSP70* gene cluster was identified in the wheat genome, which was composed of *TaHSP7-20* and *TaHSP7-21*. Of the 59 *TaHSP70* family genes, the 52 ones that accounted for 88.14% were considered segmental duplications. It can be concluded that segmental duplication played a crucial role in the *HSP70* family gene expansion, and *AetHSP70s* in *A. tauschii* were extensively homologous with *TaHSP70s* of wheat.

### 3.5. Cis-Acting Elements of the HSP70 Gene Promoter in A. tauschii

To evaluate the potential transcriptional regulation process of *AetHSP70* genes, the 2000 bp upstream promoter sequences were extracted and used to identify the *cis*-acting elements (Figure 5). CAAT-box and TATA box were two ubiquitous elements in many eukaryotic promoters; they were also predicted in all *AetHSP70s*. In addition, all promoters of *AetHSP70s* contained the presence of MeJA-responsive elements, namely the TGACG-motif and CGTCA-motif, which suggests potential involvement in the regulation of gene expression in response to methyl jasmonate signaling. *AetHSP70s* played a key role in response to MeJA. Likewise, numerous light responsiveness elements widely existed in the promoters of *AetHSP70s*, which included G-box (19), I-box (13), TCT-motif (13), GATA-motif (12), TCCC-motif (11), GTGGC-motif (8), ACE (7), GT1-motif (6), and AE-box (6). Furthermore, we found five putative motifs related to hormone response, including abscisic acid (ABRE), auxin (TGA-element and AuxRR-core), gibberellin (GARE-motif and P-box), and salicylic acid (TCA-element and SARE) (Appendix A). It was noteworthy that five *cis*-acting elements related to biotic and abiotic stress were detected as well. The anaerobic induction element (ARE) was contained in the promoter regions of 17 *AetHSP70s* except for *AetHSP70-5* and *AetHSP70-10*. More than half of the *AetHSP70s* have MYBs (15), which constituted an MYB binding site involved in drought-inducibility, and LTRs (11), a low-temperature responsiveness element. TC-rich repeats, known as defense and stress-responsive elements, were found to exist in the promoters of *AetHSP70-1*, *AetHSP70-2*, *AetHSP70-7*, *AetHSP70-16*, and *AetHSP70-17*, and a wound-responsive element (WUN-motif) was found only in *AetHSP70-7* and *AetHSP70-9*. The results suggested that *AetHSP70s* may play a regulatory role in the signaling transduction processes of stress response and tolerance for adaption to different environments.

## 4. Discussion

The *HSP70* family members, these ubiquitous molecular chaperones, are integral to a broad spectrum of cellular processes involving protein folding and remodeling. They operate throughout the entire lifespan, playing a pivotal role in preserving protein homeostasis, which has significant consequences for acclimatizing to fluctuating growth and stress conditions [38]. The *HSP70* gene family, which encodes molecular chaperones, has been identified across a diverse range of plant species, highlighting its importance in various biological processes heretofore, including *A. thaliana* [39,40], *O. sativa* [41], *Glycine max* L. [2], *Physcomitrella patens* [42], *Capsicum annuum* L. [43], and *Brassica oleracea* [44]. The *MdHSP70* gene family in apples plays a role in growth and development; and by regulating the expression of *HSP70* genes, it enhances the tolerance of apples to abiotic stress [22]. *CsHSP70* genes could be involved in how cucumbers react to hormonal signals and environmental stresses [23]. The expression of the *ZmERD2*, which encodes a member of the *HSP70* family in corn, is induced by heat, high salinity, cold, polyethylene glycol, heat stress, and dehydration treatments [24]. The *NtHSP70* family in tobacco plays a role in responding to a range of non-biological stress factors [45]. The *HSP70* genes in cotton play a regulatory role in the signaling pathways that respond to plant stress [46]. The Hsp70 protein improves the ability of mango seedlings to adapt to low temperatures [47]. The Hsp70 protein plays a regulatory role in mitigating the effects of high-temperature stress on *Porphyra yezoensis* [48]. The Hsp70-5 protein from *Pugionium cornutum* boosts the ability of genetically modified *Arabidopsis thaliana* to withstand drought by increasing the expression of genes related to stress tolerance and the activity of antioxidant enzymes [49]. The Hsp70 protein plays a regulatory role in enhancing the resilience of tomato plants against heat, drought, and salt stress [50,51]. The *clHSP70* gene in watermelon exhibits a range of responses to abscisic acid (ABA), drought conditions, and cold stress [52]. While research on the *HSP70* gene family has been conducted across various plant species, there has been a notable lack of focus on weeds. In a significant step towards understanding the genetic basis of invasive weeds, eight heat shock-related unigenes were identified in the cDNA library of Centaurea maculosa, an invasive plant species. This discovery underscores the potential role of *HSP70* genes in the adaptation and stress response mechanisms of this plant, which could be crucial for its invasive success [36]. *HSP70* and *HSP90* of *Ageratina adenophora* have been cloned and characterized to investigate the serious adaptation of this invasive alien weed [53].

In this study, we did a comprehensive genome-wide analysis of the *HSP70* gene family in *A. tauschi* using the examination of physicochemical characteristics, evolutionary relationships, structural attributes, chromosomal locations, and syntenic analysis. Based on the analysis of subcellular localization, 19 *AetHSP70s* encoding proteins were identified in four cellular compartments, including cytosol, endoplasmic reticulum, mitochondria, and plastids. It was consistent with the subcellular distribution of *HSP70s* in other plants. Within the 21 Hsp70 proteins identified in pepper, 9 members shared similar localization to the cytosol, 3 to the endoplasmic reticulum, 1 to mitochondria, 1 to the chloroplast, 1 to the plasma membrane, and 6 members were located in more than one compartment [36]. According to the phylogenetic trees of Hsp70 proteins in *A. tauschi*, wheat, Arabidopsis, and rice, the Hsp70 family members were classified into four subfamilies. Integrating with protein structure analysis, it was found that the most closely related *AetHSP70* members shared similar protein structures and motif numbers within the same subfamilies. Among the *HSP70* family members in *A. tauschi*, some genes were found to have gained introns; in other words, some *AetHSP70s* have lost introns in the coding sequence. Generally, the variation in the number and placement of intron is a common process that has occurred during evolution [54]. The factors that determine the evolutionary fate of the intron count on the intron itself, the gene in which it exists, and the host organism [55]. Interestingly, a higher number of introns in rice can lead to higher expression levels by providing post-transcriptional stability for mRNA [56].

Using the same method, we also detected 58 *TaHSP70s* in *T. aestivum*, suggesting that the abundance of *HSP70* family genes expanded and tripled, together with some duplicates, in hexaploid wheat after two polyploid ploidization events occurred [57]. The chromosomal location analysis represented both *AetHSP70s* and *TaHSP70s* are distributed across all chromosomes in *A. tauschi* and wheat, respectively. Syntenic gene, the genomic fragment analysis, which traces back to a common ancestor for various species, is primarily utilized for sharing gene annotations and unraveling the genomic evolution among related species [58]. A total of 43 pairs of syntenic *HSP70* genes were identified between *A. tauschi* and *T. aestivum*. It is recognized that individual gene duplications, segmental chromosomal duplications, and entire genome duplications have been pivotal in shaping the architecture of plant genomes. These duplication events have significantly contributed to the genetic variation, potentially fostering the emergence of novel *HSP70* functions or adaptations, enhancing stress response capabilities, and broadening the adaptability to diverse environmental conditions [59]. The previous research indicates that tandem duplications, characterized by the presence of multiple genes arrayed adjacently on the same chromosome, can be identified when two or more genes are found in close proximity. Conversely, segmental duplications are recognized by gene duplications that occur across different chromosomes, reflecting more extensive chromosomal segments involved in the duplication process. These duplication events contribute significantly to the expansion of gene families and the generation of genetic novelty within plant species [27]. In the potato genome, two pairs of *HSP70* members were identified as segmental duplication genes, and three pairs were identified as tandem duplication genes [60]. In our analyses, five *AetHSP70s* were discovered to be segmental duplications, and the 11 ones were single genes in *A. tauschi*. Hence, the segmental duplication was the main source that contributed to the expansion of the *HSP70* gene family in *A. tauschi*. For *TaHSP70* genes, besides *TaHSP70-20* and *TaHSP70-21*, which were considered tandem duplicated genes, most of the *TaHSP70s* belong to segmental duplications. After *A. tauschi* hybridized with emmer wheat to produce bread wheat, whole-genome triplication, followed by main segmental duplication and minor tandem duplication, played major roles in the expansion of the *TaHSP70* gene family. Similar expansion patterns of *HSP70s* were detected in allotetraploid *Brassica napus*, which originated from the recent genetic fusion between Brassica rapa and Brassica oleracea. This process, known as interspecific hybridization, has been pivotal in the diversification of these species, leading to a rich genetic tapestry [61,62,63]. In theory, such redundancy might offer an expanded pool of genetic diversity, enabling the emergence of new Hsp70 variants or functions, enhancing stress response efficiency, and bolstering adaptability to diverse climatic conditions [64].

*A. tauschii* has a wide geographical and environmental adaption since it spread into western China and has rapidly invaded key winter wheat-growing provinces across the whole of China. The *cis*-acting elements prediction in the *AetHSP70s* detected numerous stress-related elements involved in drought-inducibility, low-temperature responsiveness, wound-responsive elements, and hormone-related elements, including abscisic acid, auxin, gibberellin, and salicylic acid-responsive elements. These results suggested that *AetHSP70s* could be involved in a variety of stress reactions and the transmission of hormonal signals through various pathways. Hence, Hsp70 chaperones engage in stress-responsive functions, including the inhibition of protein clumping, the dissolution of protein aggregates, the facilitation of misfolded or unfolded proteins’ refolding, and the collaboration with cellular clearance systems like autophagy and the ubiquitin-proteasome pathway to eliminate abnormal proteins and aggregates [38]. Consequently, the chaperones encoded by *AetHSP70* family genes and the stress-resistant properties of *A. tauschii* are likely pivotal for its adaptability across diverse environmental conditions, potentially leading to significant ecological impacts, especially in the context of invasive plant species. These properties may underpin physiological and metabolic adaptations that enhance their competitive edge and resilience against environmental stressors.

## 5. Conclusions

This study provides a comprehensive characterization of the *HSP70* gene family in *A. tauschii*. We identified 19 *AetHSP70* genes, grouped into four subfamilies, and demonstrated extensive homology between *A. tauschii* and wheat *HSP70* genes. These results suggest evolutionary conservation and potential functional similarities. The findings highlight the role of *AetHSP70* genes in stress resistance and environmental adaptability. This work lays the foundation for future functional studies on the role of *HSP70* genes in abiotic stress tolerance. Given *A. tauschii*’s ecological impact as an invasive species, our findings could aid in developing strategies to manage its spread. Additionally, these insights may be applied to enhance stress resilience in wheat and other crops. Future research should focus on validating gene functions under specific stress conditions and exploring their applications in crop improvement and pest management.

## Figures and Tables

**Figure 1 genes-16-00019-f001:**
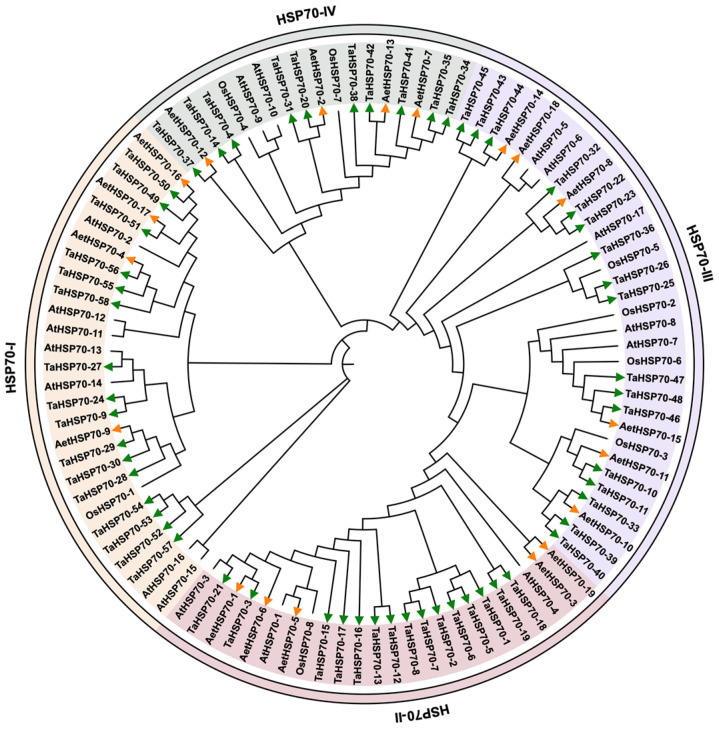
Phylogenetic analysis of Hsp70 family proteins in *A. tauschii* (AetHsp70s), *T. aestivum* (TaHsp70s), *A. thaliana* (AtHsp70s), and *O. sativa* (OsHsp70s). The complete amino acid sequences of Hsp70 proteins were aligned to identify conserved regions and divergences using the CLASTW program in MEGAX. The unrooted tree was generated using the neighbor-joining (NJ) method. All Hsp70 proteins were divided into I, II, III, and IV. Subfamilies were delineated by their unique color coding, facilitating the visual differentiation of each group. The identified AetHsp70s and TaHsp70s were marked with green and orange arrows, respectively.

**Figure 2 genes-16-00019-f002:**
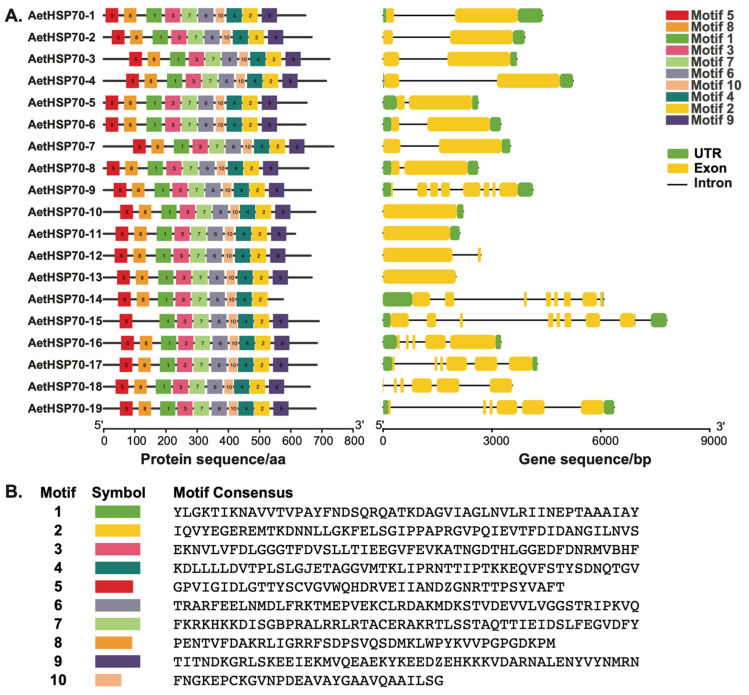
Characterizations of the identified AetHsp70s in *A. tauschii*. (**A**) In the protein structure plot (**left**), boxes with different colors denote a total of ten preserved motifs; within the gene structure diagram (**right**), the green boxes, black lines, and orange boxes correspond to regions of non-coding RNA regions (UTR), intron and exon, respectively. (**B**) Amino acid sequence composition of the conserved motif in AetHsp70 proteins.

**Figure 3 genes-16-00019-f003:**
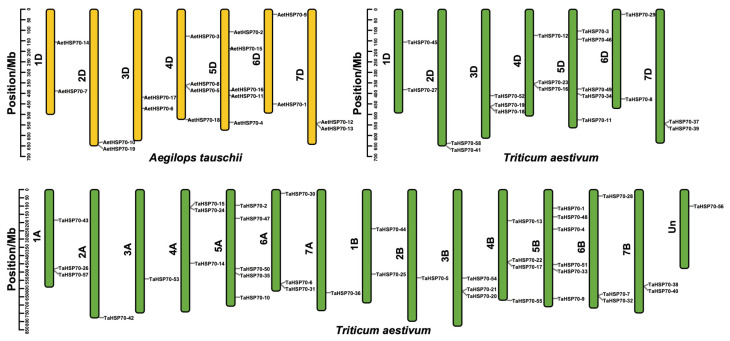
Distribution of *HSP70* family genes on *A. tauschii* and *T. aestivum* chromosomes. The yellow and green bars were represented as the chromosomes of *A. tauschii* and *T. aestivum*. The chromosome name was indicated next to each bar. Un was the unmapped chromosome of *T. aestivum*. The scale of all chromosomes was in millions of bases (Mb).

**Figure 4 genes-16-00019-f004:**
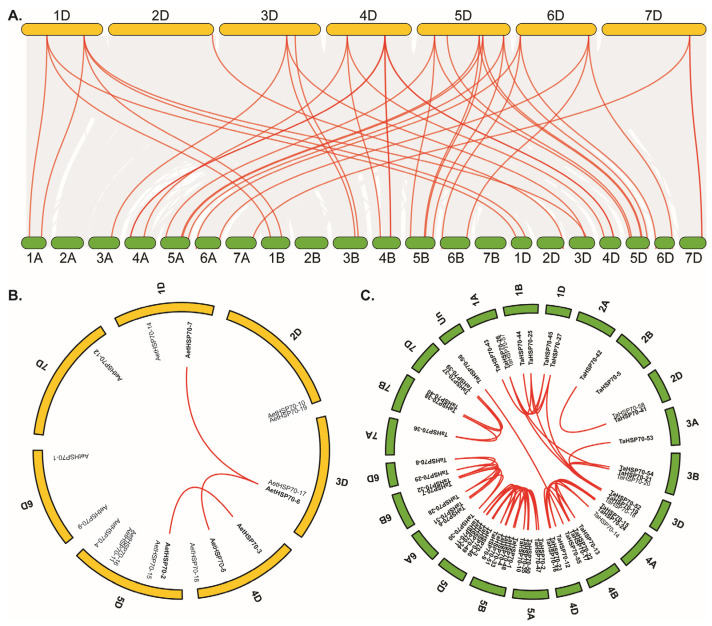
Genome-wide syntenic analysis of *HSP70* genes in *A. tauschii* and *T. aestivum*. (**A**) Synteny analysis of *HSP70* genes between *A. tauschii* and *T. aestivum*. (**B**) Synteny analysis of *HSP70* genes in *A. tauschi*. (**C**) Synteny analysis of *HSP70* genes in *T. aestivum*.

**Figure 5 genes-16-00019-f005:**
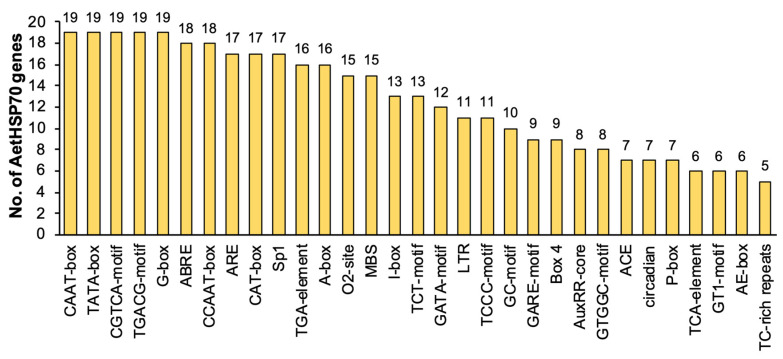
*Cis*-acting elements identified in the promoters of *AetHSP70* genes. The promoter region (−2000 bp upstream) of *AetHSP70* genes was scanned for the presence of conserved *cis*-acting elements using the PlantCare database.

**Table 1 genes-16-00019-t001:** Details of genome-wide identified *HSP 70* gene family members in *A. tauschii*.

Gene Name	Gene ID	Amino Acids	MW/kD	pI	GRAVY	Localization
*AetHSP70-1*	AET6Gv20830600	647	71.13	4.8	−0.41	Cytoplasm
*AetHSP70-2*	AET5Gv20225300	667	73.43	4.93	−0.41	ER
*AetHSP70-3*	AET4Gv20300300	724	78.93	5.19	−0.45	Plastid
*AetHSP70-4*	AET5Gv21108900	713	78.12	5.35	−0.39	Plastid
*AetHSP70-5*	AET4Gv20528300	651	71.32	4.83	−0.45	Cytoplasm
*AetHSP70-6*	AET3Gv20805300	647	70.64	4.91	−0.37	Cytoplasm
*AetHSP70-7*	AET1Gv20682100	737	81.13	7.05	−0.54	Plastid
*AetHSP70-8*	AET4Gv20525900	657	71.83	4.91	−0.44	Cytoplasm
*AetHSP70-9*	AET6Gv20108600	665	73.2	4.85	−0.47	ER
*AetHSP70-10*	AET2Gv21241200	679	73.09	4.92	−0.32	ER
*AetHSP70-11*	AET5Gv20692700	614	67.32	5.65	−0.33	ER
*AetHSP70-12*	AET7Gv21043500	664	73.11	4.87	−0.42	ER
*AetHSP70-13*	AET7Gv21044600	667	73.53	4.87	−0.39	ER
*AetHSP70-14*	AET1Gv20328300	575	61.89	5.17	−0.21	Plastid
*AetHSP70-15*	AET5Gv20312000	690	73.58	4.72	−0.27	Plastid
*AetHSP70-16*	AET5Gv20629700	684	72.74	6.04	−0.27	Mitochondrion
*AetHSP70-17*	AET3Gv20701400	683	73.34	5.57	−0.34	Mitochondrion
*AetHSP70-18*	AET4Gv20882800	661	70.74	5.08	−0.27	Mitochondrion
*AetHSP70-19*	AET2Gv21290700	680	72.85	5.05	−0.29	Mitochondrion

MW, molecular weight; pI, isoelectric point; GRAVY, grand average of hydropathicity; ER, endoplasmic reticulum.

## Data Availability

The datasets generated during the current study are available from the corresponding author upon reasonable request.

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
