# Peer review of "Genome-Wide Identification and Characterization of HSP70 Gene Family in Tausch’s Goatgrass (Aegilops tauschii)"

_genes, 2024, doi:10.3390/genes16010019_

Round 1
Reviewer 1 Report
Comments and Suggestions for Authors
Dear Authors,
The submited manuscript titled „Genome-wide identification and characterization of HSP70 gene family in Tausch's goatgrass (Aegilops tauschii)” contains interesting results. Nevertheless, I have found some mistakes, which- in my opinion- should be improved or at least clarified before an eventual publication. I have listed them below:
1. In my opinion chapter Introduction should be enlarged. I suggest to start the Intro from the current state of knowledge on Hsps (especially Hsp70) should be presented. On the basis such background the gap in knowledge will be presented and the choice of objectives of studies will be be better justified. Chapter should eneded by listing of specific goals of investigations.
2. I suggest to add short characteristics (life form, lifespan, range, etc.) of studied species as sadditional subchapter of section Material amd methods.
3. Figures 1-4 are hardly legibile. Their quality must be improved.
4. In my opinion chapter Conclusions should highlight the novelty of presented results and contain information about further directions of investigations.
5. Please, look into following publications. Perhaps some of them will be useful in manuscript improvements:
• Tkáčová, J., & Angelovičová, M. (2012). Heat shock proteins (HSPs): a review. Scientific Papers Animal Science and Biotechnologies, 45(1), 349-349.
• Lai, D. L., Yan, J., Fan, Y., Li, Y., Ruan, J. J., Wang, J. Z., ... & Cheng, J. P. (2021). Genome-wide identification and phylogenetic relationships of the Hsp70 gene family of Aegilops tauschii, wild emmer wheat (Triticum dicoccoides) and bread wheat (Triticum aestivum). 3 Biotech, 11(6), 301
• Lu, Y., Zhao, P., Zhang, A., Wang, J., & Ha, M. (2022). Genome-wide analysis of HSP70s in hexaploid wheat: tandem duplication, heat response, and regulation. Cells, 11(5), 818.
• Lai, D. L., Yan, J., Fan, Y., Li, Y., Ruan, J. J., Wang, J. Z., ... & Cheng, J. P. (2021). Genome-wide identification and phylogenetic relationships of the Hsp70 gene family of Aegilops tauschii, wild emmer wheat (Triticum dicoccoides) and bread wheat (Triticum aestivum). 3 Biotech, 11(6), 301.
• Jakhu, P., Sharma, P., Yadav, I. S., Kaur, P., Kaur, S., Chhuneja, P., & Singh, K. (2021). Cloning, expression analysis and In silico characterization of HSP101: a potential player conferring heat stress in Aegilops speltoides (Tausch) Gren. Physiology and Molecular Biology of Plants, 27, 1205-1218.
Minor remark:
The mode of incorporating literature source into a text and way of their citation in References section should be improved.
Author Response
The submited manuscript titled “Genome-wide identification and characterization of HSP70 gene family in Tausch's goatgrass (Aegilops tauschii)” contains interesting results. Nevertheless, I have found some mistakes, which in my opinion should be improved or at least clarified before an eventual publication. I have listed them below:
Comments 1: In my opinion chapter Introduction should be enlarged. I suggest to start the Intro from the current state of knowledge on Hsps (especially Hsp70) should be presented. On the basis such background the gap in knowledge will be presented and the choice of objectives of studies will be better justified. Chapter should ended by listing of specific goals of investigations.
Response 1: We appreciate the reviewer’s constructive suggestion regarding the Introduction section. Based on the feedback, we have significantly revised and expanded this section. We now start the Introduction section with a detailed overview of the current state of knowledge on heat shock proteins (Hsps), particularly focusing on the HSP70 family (Line 36-49). Following this, we have highlighted the existing gaps in knowledge regarding the HSP70 gene family in Aegilops tauschii (Line 50-64). These revisions provide a stronger justification for the study objectives, which are now explicitly listed at the end of the Introduction section (Line 65-82). We believe that these improvements enhance the clarity and logical flow of our work, as well as provide a better context for our research.
Comments 2: I suggest to add short characteristics (life form, lifespan, range, etc.) of studied species as additional subchapter of section Material and methods.
Response 2: We apologize for the omission of a description of the study species' characteristics in the Materials and Methods section. We greatly appreciate your valuable feedback, as including this information will significantly aid readers in understanding the features of the Aegilops tauschii and the significance of our research. We have now incorporated a description of the species in the Materials and Methods section (Line 86-98):
“2.1. Aegilops Tauschii Characteristics
A. tauschii, an annual grass species belonging to the Poaceae family, is considered a weed. Propagated through seeds, A. tauschii exhibits robust tillering, averaging 10-20 tillers per plant, with the maximum exceeding 32. The growth cycle of A. tauschii features two germination peaks: one in the autumn, 15-20 days after wheat sowing, and another in the spring, from late February to March of the following year. A. tauschii matures 5-7 days earlier than wheat, shedding its rachis as it matures, leaving only the base's 1-2 nodes [25]. Originating in Western Asia, A. tauschii is distributed across various provinces in China, demonstrating strong adaptability and the ability to grow in arid and saline-alkali soils. A. tauschii has a broad distribution, spanning the Eurasian continent, with Iran recognized as its center of genetic diversity and origin. In China, A. tauschii is primarily concentrated in the Xinjiang Ili region and the Yellow River basin [26].”
Comments 3:Figures 1-4 are hardly legibile. Their quality must be improved.
Response 3:We greatly appreciate your valuable feedback, which will be of immense assistance to us. There may have been a reduction in image clarity during the conversion process. We have re-uploaded images 1-5 in the latest version of the manuscript attached.
Comments 4:In my opinion chapter Conclusions should highlight the novelty of presented results and contain information about further directions of investigations.
Response 4:Thank you for your valuable feedback. Based on your suggestion, we have revised the conclusion to highlight the novelty of our findings and added potential future directions for research. The updated conclusion is as follows:
“This study provides a comprehensive characterization of the HSP70 gene family in A. tauschii. We identified 19 AetHSP70 genes, grouped into four subfamilies, and demonstrated extensive homology between A. tauschii and wheat HSP70 genes. These results suggest evolutionary conservation and potential functional similarities. The findings highlight the role of AetHSP70 genes in stress resistance and environmental adaptability. This work plays the foundation for future functional studies on the role of HSP70 genes in abiotic stress tolerance. Given A. tauschii’s ecological impact as an invasive species, our findings could aid in developing strategies to manage its spread. Additionally, these insights may be applied to enhance stress resilience in wheat and other crops. Future research should focus on validating gene functions under specific stress conditions and exploring their applications in crop improvement and pest management.” (Line 416-426)
Comments 5: Please, look into following publications. Perhaps some of them will be useful in manuscript improvements:
- Tkáčová, J., & Angelovičová, M. (2012). Heat shock proteins (HSPs): a review. Scientific Papers Animal Science and Biotechnologies, 45(1), 349-349.
- Lai, D. L., Yan, J., Fan, Y., Li, Y., Ruan, J. J., Wang, J. Z., ... & Cheng, J. P. (2021). Genome-wide identification and phylogenetic relationships of the Hsp70 gene family of Aegilops tauschii, wild emmer wheat (Triticum dicoccoides) and bread wheat (Triticum aestivum). 3 Biotech, 11(6), 301
- Lu, Y., Zhao, P., Zhang, A., Wang, J., & Ha, M. (2022). Genome-wide analysis of HSP70s in hexaploid wheat: tandem duplication, heat response, and regulation. Cells, 11(5), 818. https://doi.org/10.3390/cells11050818
- Lai, D. L., Yan, J., Fan, Y., Li, Y., Ruan, J. J., Wang, J. Z., ... & Cheng, J. P. (2021). Genome-wide identification and phylogenetic relationships of the Hsp70 gene family of Aegilops tauschii, wild emmer wheat (Triticum dicoccoides) and bread wheat (Triticum aestivum). 3 Biotech, 11(6), 301. https://doi.org/10.1007/s13205-021-02639-5
- Jakhu, P., Sharma, P., Yadav, I. S., Kaur, P., Kaur, S., Chhuneja, P., & Singh, K. (2021). Cloning, expression analysis and In silico characterization of HSP101: a potential player conferring heat stress in Aegilops speltoides (Tausch) Gren. Physiology and Molecular Biology of Plants, 27, 1205-1218. https://doi.org/10.1007/s12298-021-01005-2
Response 5:We are deeply grateful to the reviewers for providing us with several key references, which will play a significant role in improving our manuscript. The references have been added to the manuscript as appropriate, and relevant information from these sources has been integrated into the Introduction and Discussion section (Line 44-46, Line 50-52, Line 362-364, and Line 390-394).
Minor remark: The mode of incorporating literature source into a text and way of their citation in References section should be improved.
Response 6:Thank you for your valuable suggestion. In accordance with the journal's reference formatting guidelines and citation requirements, we have revised and updated the references section accordingly.

Reviewer 2 Report
Comments and Suggestions for Authors
The study was concentrated on the genome-wide identification and characterization of HSP70 gene family in Tausch's goatgrass (Aegilops tauschii). The Authors presented important results of the investigations with the proper interpretation and discussion. Among the most significant research findings are the following:
HSP70 family members were classified into four subfamilies and distributed on all chromosomes. The segmental duplication played a crucial role in HSP70 gene family expansion. Cis-acting elements prediction indicated that HSP70 family genes of A. tauschii might play essential roles in stress resistance to varied environments.
In my opinion, the manuscript may be further processed and considered for publication, after conducting some minor revisions:
- English style and grammar need to be clarified by the native speaker, specialist in the discipline (e.g. molecular biologist, bioinformatics specialist),
- Fig. 2-4: the graphical resolution should be substantially increased, because at the present form the font is poorly visible.
- In the section of Discussion, the Authors should present more information about the role of specific HSP70 genes in plants exposed to various stress conditions.
Author Response
The study was concentrated on the genome-wide identification and characterization of HSP70 gene family in Tausch's goatgrass (Aegilops tauschii). The Authors presented important results of the investigations with the proper interpretation and discussion. Among the most significant research findings are the following:
HSP70 family members were classified into four subfamilies and distributed on all chromosomes. The segmental duplication played a crucial role in HSP70 gene family expansion. Cis-acting elements prediction indicated that HSP70 family genes of A. tauschii might play essential roles in stress resistance to varied environments.
In my opinion, the manuscript may be further processed and considered for publication, after conducting some minor revisions:
Comments 1: English style and grammar need to be clarified by the native speaker, specialist in the discipline (e.g. molecular biologist, bioinformatics specialist).
Response 1: We appreciate the reviewer’s comment. We have enlisted the expertise of a bioinformatics specialist to refine the language and grammar of our manuscript. The revised draft has been re-uploaded to the attachment.
Comments 2: Fig. 2-4: the graphical resolution should be substantially increased, because at the present form the font is poorly visible.
Response 2: Thank you for pointing this out. There may have been a reduction in image clarity during the conversion process. We have re-uploaded all the Figures 1-5 in the latest version of the manuscript attached.
Comments 3: In the section of Discussion, the Authors should present more information about the role of specific HSP70 genes in plants exposed to various stress conditions.
Response 3: We greatly appreciate your valuable feedback. We all agree that a more detailed analysis of the specific functions of HSP70 genes in stress responses would enrich the manuscript. Hence, we included additional information in the Discussion section to address the roles of specific HSP70 genes in plants under different stress conditions as below:
“The MdHSP70 gene family in apples plays a role in growth and development, and by regulating the expression of HSP70 genes, it enhances the tolerance of apples to abiotic stress [22]. CsHSP70 genes could be involved in how cucumbers react to hormonal signals and environmental stresses [23]. The expression of the ZmERD2, which encodes a member of the Hsp70 family in corn, is induced by heat, high salinity, cold, polyethylene glycol, heat stress, and dehydration treatments [24]. The NtHsp70 family in tobacco plays a role in responding to a range of non-biological stress factors [47]. The HSP70 genes in cotton play a regulatory role in the signaling pathways that respond to plant stress [48]. The Hsp70 protein improves the ability of mango seedlings to adapt to low temperatures [49]. The Hsp70 protein plays a regulatory role in mitigating the effects of high-temperature stress on Porphyra yezoensis [50]. The Hsp70-5 protein from Pugionium cornutum boosts the ability of genetically modified Arabidopsis thaliana to withstand drought by increasing the expression of genes related to stress tolerance and the activity of antioxidant enzymes [51]. The Hsp70 protein plays a regulatory role in enhancing the resilience of tomato plants against heat, drought, and salt stress [52, 53]. The clHSP70 gene in watermelon exhibits a range of responses to abscisic acid (ABA), drought conditions, and cold stress [54].” (Line 317-333)

Reviewer 3 Report
Comments and Suggestions for Authors
The manuscript aims to identify and characterize the HSP70 gene family in Aegilops tauschii to understand its role in stress resistance and environmental adaptation. Using genome-wide analysis, the authors identified 19 HSP70 genes and analyzed their physicochemical properties, phylogenetic relationships, gene structures, chromosomal distributions, and cis-acting regulatory elements. The study contributes to understanding the molecular basis of stress resistance in invasive weeds and establishes synteny with wheat.
-The genome-wide identification appears robust, utilizing both BLAST and HMM searches. However, additional functional validation, such as transcriptomic or proteomic approaches, would strengthen the findings.
-How did you validate the predictions for subcellular localization and cis-acting elements? Were any experimental methods, such as qPCR or reporter assays, used to support these bioinformatic findings?
-Could you explain why segmental duplications were considered the primary driver of HSP70 family expansion in A. tauschii? How do these results compare with duplication patterns in other species?
-What do you hypothesize about the absence of certain motifs (e.g., motifs 8 and 9) in AetHSP70-14 and AetHSP70-15? How might this influence their functional roles?
-How do you see these findings applied practically in managing invasive species or improving crop resilience?
-Do you foresee any challenges leveraging this genetic information to control A. tauschii or enhance wheat resilience?
-In Figure 5, how did you prioritize which cis-acting elements to highlight? Are there additional elements that might warrant discussion?
Author Response
The manuscript aims to identify and characterize the HSP70 gene family in Aegilops tauschii to understand its role in stress resistance and environmental adaptation. Using genome-wide analysis, the authors identified 19 HSP70 genes and analyzed their physicochemical properties, phylogenetic relationships, gene structures, chromosomal distributions, and cis-acting regulatory elements. The study contributes to understanding the molecular basis of stress resistance in invasive weeds and establishes synteny with wheat.
Comments 1: The genome-wide identification appears robust, utilizing both BLAST and HMM searches. However, additional functional validation, such as transcriptomic or proteomic approaches, would strengthen the findings.
Response 1: Thank you very much for your valuable feedback. We highly appreciate your suggestion regarding the addition of functional validation through transcriptomic or proteomic approaches. We completely agree that such validation would greatly enhance the robustness of our findings. However, as this study is based on bioinformatics analysis using genomic data from Aegilops tauschii and wheat, and due to the current unavailability of the Aegilops tauschii samples in our laboratory, we were unable to detect transcriptomic and proteomic information to validate our results. We fully recognize the importance of this aspect and plan to incorporate transcriptomic or proteomic validation in future experiments, as you have suggested. We will certainly take your recommendation into account to strengthen the study in subsequent phases of our research.
Comments 2: How did you validate the predictions for subcellular localization and cis-acting elements? Were any experimental methods, such as qPCR or reporter assays, used to support these bioinformatic findings?
Response 2: Thank you for your thoughtful question regarding the validation of our predictions for subcellular localization and cis-acting elements. As mentioned above that this study was primarily based on bioinformatic analysis. The predictions for subcellular localization were made using prediction tool DeepLoc (DeepLoc: Prediction of protein subcellular localization using deep learning), which provide reliable in silico results based on sequence data. Similarly, cis-acting elements were identified through the analysis of upstream regulatory regions using online databases PlantCARE (https://bioinformatics.psb.ugent.be/webtools/plantcare/html/), which are widely accepted for their accuracy in predicting potential regulatory elements.
However, we acknowledge that experimental validation, such as qPCR or reporter assays, would provide more direct evidence to support these predictions. Due to the current focus of our study on genome-wide identification and characterization, we did not perform these experimental validations. We will address this gap in future experiments by performing qPCR and potentially reporter assays to validate the predicted subcellular localization and cis-acting elements, as these methods will offer more concrete evidence to complement our bioinformatic findings.
Comments 3: Could you explain why segmental duplications were considered the primary driver of HSP70 family expansion in A. tauschii? How do these results compare with duplication patterns in other species?
Response 3: We greatly appreciate your insightful comment. Segmental duplication is considered the primary driver of HSP70 family expansion in Aegilops tauschii because it leads to the multiplication of genomic fragments across different chromosomes, thereby increasing the number of HSP70 genes.
In comparison with other species, previous studies have shown that segmental duplications are a common mechanism driving the expansion of multigene families, including HSP70, especially in plants. For instance, in wheat (Triticum aestivum), a close relative of Aegilops tauschii segmental duplications have also been reported as a major factor in the expansion of the HSP70 family [59]. Similarly, in rice (Oryza sativa), segmental duplications, along with tandem duplications, have been recognized as key contributors to the diversification and expansion of gene families involved in stress responses [43]. In the potato genome, two pairs of HSP70 members were identified as segmental duplication genes, and three pairs were identified as tandem duplication genes [62]. Moreover, the expansion pattern of HSP70s in Aegilops tauschii is similar to that observed in allotetraploid Brassica napus, which also experienced segmental duplications following interspecific hybridization between Brassica rapa and Brassica oleracea. Comparative analysis revealed that whole-genome triplication and segmental duplication events are the primary drivers of expansion for the HSP70 gene family within the Brassica genus [65].
Comments 4: What do you hypothesize about the absence of certain motifs (e.g., motifs 8 and 9) in AetHSP70-14 and AetHSP70-15? How might this influence their functional roles?
Response 4: We greatly appreciate your kind question. In this study, we have not hypothesized the specific functional implications of the absence of certain motifs, such as motifs 9 and 8, in AetHSP70-14 and AetHSP70-15. However, we believe that the absence of these motifs may indicate functional divergence within the HSP70 gene family in Aegilops tauschii. Motifs in protein sequences are often associated with specific functional roles, including substrate recognition or interaction with co-chaperones. The lack of these motifs in AetHSP70-14 and AetHSP70-15 could potentially result in altered substrate specificity, changes in their protein-protein interaction networks, or distinct roles in stress response pathways compared to other members of the HSP70 family. Further experimental studies would be needed to clarify the exact functional implications of these differences.
Comments 5: How do you see these findings applied practically in managing invasive species or improving crop resilience?
Response 5: Thank you for raising this meaningful question. The findings from the present study could provide some theoretical basis for both managing invasive species like Aegilops tauschii and improving crop resilience. Our identification and characterization of the HSP70 gene family in Aegilops tauschii may inform strategies to control its spread in wheat-growing regions. Since HSP70 genes play a crucial role in stress resistance, targeting these genes could potentially disrupt the weed's ability to tolerate harsh environments, thereby reducing its invasiveness. By understanding the stress-related functions of HSP70 genes in Aegilops tauschii, we could identify potential candidates for gene transfer or gene editing to improve the resistance of wheat against abiotic stresses. This may offer useful information to develop more resilient crops, which are essential for maintaining food security in the face of climate change.
Comments 6: Do you foresee any challenges leveraging this genetic information to control A. tauschii or enhance wheat resilience?
Response 6: We sincerely appreciate the valuable feedback you have provided for our manuscript. While the genetic information gained from this study provides valuable insights for managing Aegilops tauschii and enhancing wheat resilience, there are many challenges to leveraging this information in practice. One potential challenge is the complexity of gene function in the context of the whole organism. While we have known HSP70 family genes involved in stress responses, the actual role of these genes in different environmental conditions and stages of growth may require further investigation. Additionally, Insufficient functional research on the HSP70 gene family and the complexity of stress breeding in wheat require further study. The intricate regulation of HSP70 gene expression under various stressors necessitates deeper investigation into their roles in plant stress resistance.
Comments 7: In Figure 5, how did you prioritize which cis-acting elements to highlight? Are there additional elements that might warrant discussion?
Response 7: Thank you for your thoughtful question regarding Figure 5. In our study, we emphasize cis-acting elements because they are a core component of promoters, playing a crucial role in regulating the transcription process and significantly affecting promoter activity. Different promoter elements have their unique sequence structural features and functions, for example, the TATA-box and CAAT-box are universally present and essential promoter elements in eukaryotes. Furthermore, aside from cis-acting elements, factors such as the diversity of promoter activity, response to environmental stress, complexity of gene expression, universality of promoters, and regulatory mechanisms of promoters should also be considered. These factors collectively influence the expression and function of Aegilops tauschii HSP70 genes and are essential for understanding their regulatory networks within organisms.

Reviewer 4 Report
Comments and Suggestions for Authors
The manuscript under evaluation contains quite interesting results of the HSP70 family genes of Aegilops tauschii. These genes may play an important role in stress resistance. It is important to note that these genes are largely homologous between Aegilops tauschii and wheat. Can the research results obtained be used in breeding more resistant wheat varieties to various stress conditions? In what time frame would this be possible? What obstacles stand in the way?
Comments
Materials and Methods
Where was the research done? In which year?
Please provide full details of the producer of the statistical software used
References
Please limit your selection of publications to the most recent.
Author Response
The manuscript under evaluation contains quite interesting results of the HSP70 family genes of Aegilops tauschii. These genes may play an important role in stress resistance. It is important to note that these genes are largely homologous between Aegilops tauschii and wheat. Can the research results obtained be used in breeding more resistant wheat varieties to various stress conditions? In what time frame would this be possible? What obstacles stand in the way?
Response: We would like to sincerely thank you for your valuable feedback on our manuscript. We fully agree with your observation that the HSP70 gene family in Aegilops tauschii plays a crucial role in stress resistance. Its homology with wheat highlights the significant potential for enhancing wheat resilience.
Regarding the potential application of our findings in breeding more stress-resistant wheat varieties, we believe this is indeed feasible. Further research into the function of these genes in wheat could help identify key genes associated with stress tolerance, which could then be incorporated into molecular breeding programs. Specifically, molecular markers based on the HSP70 genes could serve as effective tools for selecting wheat lines with enhanced stress resistance.
However, several challenges must be addressed before these findings can be translated into practical breeding applications. First, the functional research on the HSP70 gene family in wheat is still limited, and the specific functions and mechanisms of these genes need further validation through additional experiments. Additionally, wheat’s complex genome and the polygenic nature of stress resistance traits present significant obstacles. Moreover, the regulation of HSP70 gene expression under various stress conditions is highly intricate, and further investigation is needed to better understand its role in plant stress resistance.
Despite these challenges, we are optimistic about the potential of using the HSP70 genes identified in Aegilops tauschii to improve wheat stress resistance. Achieving this will require sustained effort over an extended period to overcome genetic, breeding, and environmental complexities associated with wheat improvement. Further research and experimental validation will be crucial to fully realize the practical benefits of these findings in breeding programs.
Comments 1:
Materials and Methods
Where was the research done? In which year?
Please provide full details of the producer of the statistical software used.
Response 1: We sincerely thank the reviewer for pointing out the need for clarification. The research presented in this paper primarily focuses on the bioinformatics analysis of the HSP70 gene family in Aegilops tauschii. The majority of the study was conducted at Qingdao Agricultural University, and the research was carried out between 2023 and 2024.
Regarding the statistical software used, we have provided the relevant reference for the software or the link to the database in the Materials and Methods section. The specific details of the software are appropriately cited in the References section. We hope this resolves the query.
Comments 2: References
Please limit your selection of publications to the most recent.
Response 2: We sincerely appreciate the valuable feedback you have provided on our manuscript. We have carefully reviewed the references and have updated the manuscript to include more recent publications in the References section as below:
1 Huang, Y. C., Liu, C. C., Li, Y. J., Liao, C. M., Vivek, S ., Chuo, G. L., Tseng, C. Y., Wu, Z. Q., Shimada, T., Suetsugu, N., Wada, M., Lee, C. M., Jinn, T. L. 2024. Multifaceted roles of Arabidopsis heat shock factor binding protein in plant growth, development, and heat shock response. Environmental and Experimental Botany, 226, 105878. https://doi.org/10.1016/j.envexpbot.2024.105878
21 Wang, N., Chen, H. 2024. Effects of Soil Drought on Competitiveness of the Invasive Weed Aegilops tauschii. Russian Journal of Plant Physiology, 71,114. https://doi.org/10.21203/rs.3.rs-4311260/v1
22 Liu, M., Bian, Z. Y., Shao, M., Feng, Y. Q., Ma, W. F., Liang, G. P., Mao, J. 2024.Expression analysis of the apple HSP70 gene family in abiotic stress and phytohormones and expression validation of candidate MdHSP70 genes. Scientific Reports, 14:23975. https://doi.org/10.1038/s41598-024-73368-x
23 Zhou, Z. X., Xiao, L. D., Zhao, J. D., Hu, Z. Y., Zhou, Y. L., Liu, S. Q., Zhou Y. 2023.Comprehensive Genomic Analysis and Expression Profile of Hsp70 Gene Family Related to Abiotic and Biotic Stress in Cucumber. Horticulturae, 9:1057. https://doi.org/10.3390/horticulturae9091057
49 Huang, Y. X., Chen, M. M., Chen, D. M., Chen, H. M., Xie, Z. H., Dai, S. F. 2024. Enhanced HSP70 binding to m6A-methylated RNAs facilitates cold stress adaptation in mango seedlings. BMC Plant Biology, 24: 1114. https://doi.org/10.1186/s12870-024-05818-7
50 Huang, D. L., Tian, C., Sun, Z. J., Niu, J. F., Wang, G. C. 2024. Potential synergistic regulation of hsp70 and antioxidant enzyme genes in Pyropia yezoensis under high temperature stress. Algal Research, 78: 103375. https://doi.org/10.1016/j.algal.2023.103375
51 Xu, K., Wang, P. 2024.Transcriptome‑wide identification of the Hsp70 gene family in Pugionium cornutum and functional analysis of PcHsp70‑5 under drought stress. Planta, 260: 84. https://doi.org/10.1007/s00425-024-04509-9
52 Xu, T., Zhou, H., Feng, J., Guo, M. Y., Huang, H. M., Yang, P., Zhou J. 2024.Involvement of HSP70 in BAG9-mediated thermotolerance in Solanum lycopersicum. PlantPhysiologyandBiochemistry, 207: 108353. https://doi.org/10.1016/j.plaphy.2024.108353
53 Vu, N. T., Nguyen, N. B .T., Ha, H. H., Nguyen, L. N., Luu, L. H., Dao, H. Q., Vu, T. T., Huynh, H. T. T., Le H. T. T. 2023. Evolutionary analysis and expression profiling of the HSP70 gene family in response to abiotic stresses in tomato (Solanum lycopersicum). Science Progress, 106(1):368504221148843. https://doi.org/10.1177/00368504221148843
54 Wang, X. S., Jin, Z., Ding, Y., Guo, M. 2023. Characterization of HSP70 family in watermelon (Citrullus lanatus): identification, structure, evolution, and potential function in response to ABA, cold and drought stress. Genet, 14:1201535. https://doi.org/10.3389/fgene.2023.1201535
